# Migratory network reveals unique spatial-temporal migration dynamics of Dunlin subspecies along the East Asian-Australasian Flyway

Benjamin J. Lagassé[1¤a]*, Richard B. Lanctot[2], Stephen Brown[3], Alexei G. Dondua[4], Steve Kendall[5], Christopher J. Latty[5], Joseph R. Liebezeit[6¤b], Egor Y. Loktionov[7], Konstantin S. Maslovsky[8], Alexander I. Matsyna[9], Ekaterina L. Matsyna[9], Rebecca L. McGuire[10], David C. Payer[5¤c], Sarah T. Saalfeld[2], Jonathan C. Slaght[10], Diana V. Solovyeva[11], Pavel S. Tomkovich[12], Olga P. Valchuk[8], Michael B. Wunder[1]

1 Department of Integrative Biology, University of Colorado Denver, Denver, CO, United States of America, 2 Division of Migratory Bird Management, U.S. Fish and Wildlife Service, Anchorage, AK, United States of America, 3 Manomet, Inc., Saxtons River, VT, United States of America, 4 Beringia National Park, Providenia, Russia, 5 Arctic National Wildlife Refuge, U.S. Fish and Wildlife Service, Fairbanks, AK, United States of America, 6 Wildlife Conservation Society, Portland, OR, United States of America, 7 Bauman Moscow State Technical University, Moscow, Russia, 8 Federal Scientific Center of the East Asia Terrestrial Biodiversity, Far Eastern Branch of the Russian Academy of Sciences, Vladivostok, Russia, 9 Working Group on Waders of Northern Eurasia, Nizhny Novgorod, Russia, 10 Arctic Beringia Regional Program, Wildlife Conservation Society, Fairbanks, AK, United States of America, 11 Institute of Biological Problems of the North, Magadan, Russia, 12 Zoological Museum, Lomonosov Moscow State University, Moscow, Russia

¤a Current address: Department of Biology and Wildlife, University of Alaska Fairbanks, Fairbanks, AK, United States of America
¤b Current address: Portland Audubon, Portland, OR, United States of America
¤c Current address: National Park Service, Anchorage, AK, United States of America
* benjamin.j.lagasse@gmail.com

## Abstract

Determining the dynamics of where and when individuals occur is necessary to understand population declines and identify critical areas for populations of conservation concern. However, there are few examples where a spatially and temporally explicit model has been used to evaluate the migratory dynamics of a bird population across its entire annual cycle. We used geolocator-derived migration tracks of 84 Dunlin (*Calidris alpina*) on the East Asian-Australasian Flyway (EAAF) to construct a migratory network describing annual subspecies-specific migration patterns in space and time. We found that Dunlin subspecies exhibited unique patterns of spatial and temporal flyway use. Spatially, *C. a. arcticola* predominated in regions along the eastern edge of the flyway (e.g., western Alaska and central Japan), whereas *C. a. sakhalina* predominated in regions along the western edge of the flyway (e.g., N China and inland China). No individual Dunlin that wintered in Japan also wintered in the Yellow Sea, China seas, or inland China, and vice-versa. However, similar proportions of the 4 subspecies used many of the same regions at the center of the flyway (e.g., N Sakhalin Island and the Yellow Sea). Temporally, Dunlin subspecies staggered their south migrations and exhibited little temporal overlap among subspecies within shared migration regions. In contrast, Dunlin subspecies migrated simultaneously during north

**Data Availability Statement:** All relevant data are within the paper and its Supporting Information files.

**Funding:** This work was funded by the American Ornithological Society–Alexander Wetmore Research Award, NGO Amur–Ussuri Center for Avian Biodiversity (Vladivostok, Russia), Arctic Landscape Conservation Cooperative, Arctic Shorebird Demographics Network, BirdsRussia, Bureau of Land Management, Calvin J. Lensink Fund, The MacArthur Foundation, Manomet Inc.–Shorebird Recovery Program, National Fish and Wildlife Foundation, Neotropical Migratory Bird Conservation Act Grant Program, The Nuttall Ornithological Club–Blake-Nuttall Fund Grant, The Trust for Mutual Understanding, U.S. Fish and Wildlife Service (Avian Influenza Program, Migratory Bird Management Division, National Wildlife Refuge Challenge Cost Share Program, National Wildlife Refuge Division), Wildlife Conservation Society–Arctic Beringia Regional Program, and the Wilson Ornithological Society–Paul A. Stewart Grant. The funders had no role in study design, data collection and analysis, decision to publish, or preparation of the manuscript.

**Competing interests:** The authors have declared that no competing interests exist.

migration. South migration was also characterized by individuals stopping more often and for more days than during north migration. Taken together, these spatial-temporal migration dynamics indicate Dunlin subspecies may be differentially affected by regional habitat change and population declines according to where and when they occur. We suggest that the migration dynamics presented here are useful for guiding on-the-ground survey efforts to quantify subspecies' use of specific sites, and to estimate subspecies' population sizes and long-term trends. Such studies would significantly advance our understanding of Dunlin space-time dynamics and the coordination of Dunlin conservation actions across the EAAF.

## Introduction

For millions of Arctic-breeding shorebirds, seasonal migrations span thousands of kilometers and present survival risks that can in turn affect future productivity, and thus the growth trajectory of a population [1–3]. Determining the extent to which individuals are spatially and temporally connected to particular flyway areas is, therefore, an important component of coordinated conservation strategies designed to halt or reverse population declines [4–7]. For small Arctic-breeding shorebirds (< 100 g), archival tracking devices facilitate the estimation of population-level migratory connectivity [5, 8] between breeding, migration, and wintering areas, and are a useful tool for studying drivers of population declines [9, 10]. Studies employing archival tracking devices often focus specifically on understanding spatial and temporal connectivity between breeding and wintering areas [9–12]. However, understanding the spatial and temporal dynamics of flyway areas that connect breeding and wintering areas is also important for determining drivers and consequences of population declines [13–15].

Constructing a migratory network from animal tracking data is one approach to assess spatial and temporal connections among flyway areas and the migrants that use them [16–19]. Adapted from network theory, a migratory network combines movement data from multiple individuals to graphically summarize how breeding and nonbreeding areas (i.e., network nodes) are interconnected via immigration and emigration (i.e., network edges; [20]. Network nodes and edges may also be characterized (i.e., weighted) by the quantity of individuals that use them. Once constructed, a migratory network is a powerful framework for evaluating how populations use a flyway in space and time [13, 21], estimating the tradeoffs associated with various conservation actions [6, 7], and helping to understand drivers of population trends [16, 22]. However, there are comparatively few examples where spatially and temporally explicit migratory networks have been constructed to evaluate the migration dynamics of a bird population across its entire annual cycle [e.g., 18, 19, 23].

### Study system

The East Asian-Australasian Flyway (EAAF) has the greatest proportion of threatened and near-threatened migratory bird species of any global flyway [24]. At least 60% of Arctic-breeding shorebird populations have declined by up to 8% annually [2, 25], and the available evidence suggests declines are primarily linked to anthropogenic habitat degradation at key stopping and wintering sites in the Yellow Sea [2, 26–28]. However, for many species on the EAAF, prioritizing flyway conservation actions has been difficult due to limited information regarding population migration patterns [29–31].

The Dunlin (*Calidris alpina*) is a species of sandpiper that spends the winter in coastal and interior wetlands of East Asia and migrates to Arctic and sub-Arctic breeding areas in eastern

Russia and northern Alaska. Within the Beringia breeding region, there are 5 recognized subspecies. Each subspecies has their own distinct breeding range (Fig 1), and population sizes vary from ~2,000–500,000 [32, 33]. Dunlin that breed in northern Alaska (*C. a. arcticola*) appear to have concerningly low adult survival rates (*S* = 0.54; [34]), and surveys indicate Dunlin populations have likely declined in the Republic of Korea (H.-J. Kim, pers. comm.), the People's Republic of China (28% from 1996–2014; [35; but see 36]), and Japan (up to 80% from 1975–2008; [37]). However, prioritizing flyway conservation actions for specific populations has been difficult because Dunlin subspecies are visually indistinguishable in the field and, therefore, we have an incomplete understanding of where and when on the EAAF each subspecies occurs [33, 38, 39].

By compiling band recoveries, Lagassé et al. [40] provided the first detailed information on the migration patterns of the 4 Dunlin subspecies that migrate and winter along the EAAF (*C. a. actites*, *arcticola*, *kistchinski*, *and sakhalina*). In their analysis, the authors found that Dunlin in Japan are predominantly *C. a. arcticola*, while Dunlin migrating and wintering in other areas of the EAAF may comprise all 4 subspecies. The authors also found that continued habitat degradation at key sites in the Yellow Sea would likely have a strong negative effect on all 4 Dunlin subspecies, because 21–50% of subspecific migration recoveries were connected to the region. Although these findings are a useful step toward understanding Dunlin migration dynamics on the EAAF, Lagassé et al. [40] failed to locate any recoveries of the *kistchinski* subspecies and warned of the likely effects of regionally biased observer effort on regional recovery patterns. The authors were also unable to determine how birds moved between initial capture sites and recovery sites, and consequently, lacked information on the spatial and temporal migration dynamics of the 4 subspecies. These knowledge gaps continue to impede identification of the network of sites visited by Dunlin subspecies, their relative use, and therefore, the prioritization of flyway conservation actions for declining Dunlin populations on the EAAF [7, 38, 41].

Here, we aggregate geolocator-derived migration tracks of 84 Dunlin and construct a migratory network to evaluate the spatial and temporal migration dynamics of the 4 Dunlin subspecies that migrate and winter along the EAAF. Consistent with earlier work on Dunlin migration patterns in the western Palearctic [42, 43], we predicted the distribution of Dunlin subspecies along the EAAF would reflect the geographic distribution of the subspecies on their breeding grounds (i.e., exhibit a parallel migration pattern; Fig 1). For example, we expected *C. a. arcticola*, which breeds farthest north and east on the flyway (i.e., Alaska), to winter farthest north and east (i.e., Japan). Similarly, we expected *C. a. actites*, which breeds farthest south and west on the flyway (i.e., Sakhalin Island), to winter farthest south and west (i.e., the South China Sea). Following this pattern, we expected *C. a. sakhalina* and *kistchinski*, which have breeding ranges between the 2 other subspecies, to winter in intermediate regions, such as the Yellow Sea and East China Sea, respectively. Finally, following Holmes [44] and Tomkovich [45], we predicted Dunlin subspecies would migrate asynchronously, with subspecies breeding farther north departing/arriving on breeding, migration, and wintering grounds later, and subspecies that breed farther south departing/arriving earlier. We expected to find this pattern because spring phenology is later at northern breeding sites used by the 4 subspecies, and because Dunlin generally breed once favorable conditions become available in the spring and migrate to intertidal staging areas soon after breeding is complete [44, 45].

## Methods

### Geolocator deployment and recapture

Nests were located at 8 Arctic and sub-Arctic breeding sites using systematic area searches or by rope dragging suitable habitats (Table 1 and Fig 1; [46]). Adults were captured at nests

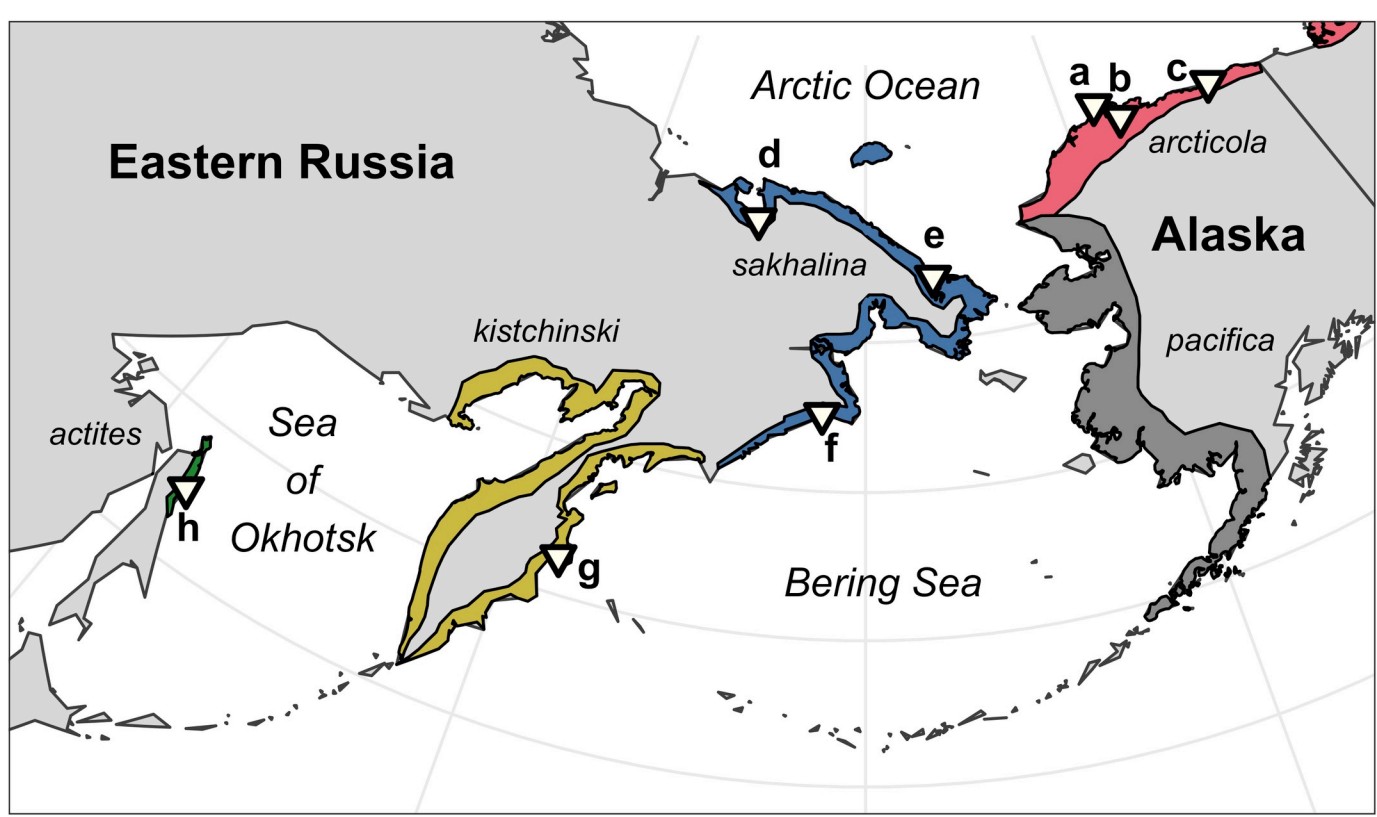

**Fig 1.** Breeding ranges of the 5 Dunlin subspecies that occur in the North Pacific, and the location of each field site (a–h) where light-level geolocators were deployed. See Table 1 for site info. The *pacifica* subspecies does not migrate and winter along the East Asian-Australasian Flyway and is not discussed in this paper.

**Table 1. Location, subspecies, and number of Dunlin equipped and later recaptured with light-level geolocators at 8 field sites along the East Asian-Australasian Flyway.** Site locations are in Fig 1.

| Site id | Field site | Latitude, longitude | Subspecies | Deployment–recapture year | Total deployed | Geolocator model | Total recovered[a]C \| P |
|---|---|---|---|---|---|---|---|
| a | Utqiaġvik, Alaska | 71.2652, -156.6359 | *arcticola* | 2010–11 | 51 | Mk12 | 15 \| 3 |
| | | | | 2016–17 | 46 | Intigeo-W65 | 16 \| 0 |
| | | | | 2017–18 | 8 | Intigeo-W65 | 1 \| 0 |
| | | | | 2018–19 | 40 | Intigeo-W65 | 10 \| 1 |
| b | Ikpikpuk River, Alaska | 70.5525, -154.7309 | *arcticola* | 2010–11 | 35 | Mk12 | 4 \| 1 |
| c | Canning River, Alaska | 70.1180, -145.8506 | *arcticola* | 2010–11 | 22 | Mk12 | 3 \| 2 |
| | | | | 2016–17 | 13 | Intigeo-W65 | 3 \| 0 |
| d | Chaun Delta, Russia | 68.7750, 170.5495 | *sakhalina* | 2013–14 | 35 | Intigeo-W65 | 4 \| 6 |
| e | Belyaka Spit, Russia | 67.0647, -174.5000 | *sakhalina* | 2011–12 | 10 | Mk12 | 4 \| 1 |
| | | | | 2013–14 | 15 | Intigeo-W65 | 5 \| 1 |
| | | | | 2016–17 | 14 | Intigeo-W65 | 6 \| 1 |
| f | Meinypilgyno, Russia | 62.5833, 177.0300 | *sakhalina* | 2014–15 | 5 | Intigeo-W65 | 3 \| 0 |
| | | | | 2016–17 | 7 | Intigeo-W65 | 4 \| 0 |
| G | Cape Pogodny, Russia | 56.2645, 162.5815 | *kistchinski* | 2017–18 | 20 | Intigeo-W65 | 5 \| 0 |
| H | Chaivo Bay, Russia | 52.5000, 143.2833 | *actites* | 2016–17 | 18 | Intigeo-W65 | 1 \| 0 |

[a]C = number of tags with complete migration tracks, P = number with partial migration tracks.

during incubation or while attending broods using a bow net or mist net, respectively [47]. Once captured, we attached a unique metal band, a Darvic leg flag with an affixed light-level geolocator, and a unique combination of color bands to the tibiotarsus and tarsometatarsus. The total weight of the leg flag with affixed geolocator was < 3% of the mean body mass of the smallest subspecies [48]. Although our tags were within suggested weight limits [49], leg-mounted geolocators were found to decrease individual return rates by 43% at 2 of 3 sites where *C. a. arcticola* were tagged in 2010 (Table 1; [50]). However, the same analysis found return rates were unchanged in 3 other Dunlin subspecies (*C. a. hudsonia*, *pacifica, and schinzii*), and the authors suggested that future studies could mitigate impacts of tags by minimizing the use of additional markers [50]. We minimized the use of additional markers after 2010, and tags were recovered from individuals that successfully migrated and exhibited typical breeding behavior the following year (i.e., as they incubated eggs or attended broods); suggesting that carrying a tag likely did not alter the behavior of the birds presented here. Geolocators were placed on both adults of a breeding pair, when possible, to maintain an even ratio of males and females within our tagged populations. Adults were also measured (wing, tarsometatarsus, total head, culmen) and had feathers and/or blood collected for archival purposes [51].

**Ethics statement.**   Permits to capture and tag *C. a. arcticola* were approved by the U.S. Geological Survey Bird Banding Laboratory (permit 23269, 23566), the U.S. Fish and Wildlife Service (permit MB-085371), the Alaska Department of Fish and Game (permit 10–044, 11–018, 16–111, 17–102, 18–160, 19–154), the North Slope Borough, and the Ukpeaġvik Iñupiat Corporation. Trapping and handling procedures were carried out in accordance with Institutional Animal Care and Use Committee protocols (Bishop's University BUACC 2009–07; U.S. Fish and Wildlife Service Alaska Region IACUC 2016–005 and 2019–008). Within Russia, permits were not required to capture and tag *C. a. kistchinski* or *C. a. sakhalina*. Trapping and handling of *C. a. actites* was approved by the Russian Federal Service for Supervision of Natural Resources (permit 2016–62, 2017–42).

## Geolocator analysis

We used 2 models of light-level geolocators (Mk12, British Antarctic Survey, and Intigeo-W65, Migrate Technology Inc.) across the 9-year study period (2010–2019; Table 1). Geolocators measured ambient light levels every minute and recorded the maximum value across either a 2- (Mk12) or 5-minute (W65) sampling period. Mk12 geolocators truncated the maximum light value on an arbitrary scale from 0–64, and therefore, only detected coarse-scale changes in ambient light levels (e.g., dusk and dawn). In contrast, W65 geolocators did not truncate light values and detected fine-scale changes in ambient light levels throughout each day. Ambient light levels were downloaded from recovered tags and any offset between the geolocator's internal clock and Greenwich Mean Time (i.e., clock drift) was linearly corrected [52]. We were unable to address potential clock drift for tags that were non-functional upon recovery (16 of 100 tags; Table 1).

Migration tracks were generated from ambient light level readings using the TwGeos (v0.0–1; [53]) and FLightR (v0.4.9; [54]) packages in program R (v3.5.2; [55]). First, we used the findTwilights function to identify the time of each sunrise and sunset (i.e., twilights; [56]). These twilights were defined as the Greenwich Mean Time when light levels crossed an arbitrary threshold of 5 (Mk12) or 12.5 lux (W65). We then used the twilightEdit function to identify and discard incorrect twilight assignments due to periodic shading of the light sensor (e.g., if an adult roosted with the geolocator tucked among its body feathers). A twilight was considered incorrect and discarded if it was > 45 minutes different than the corresponding twilights

that occurred in the surrounding 4 days (2 days before and 2 days after; [56]). Second, we calibrated individual geolocators using twilight data collected at a known location before geolocators were deployed or after they were recovered (i.e., "rooftop" calibration), or while geolocators were attached to birds on their breeding grounds (i.e., "in-habitat" calibration; [57]). We preferentially used an in-habitat calibration except at breeding sites north of 66.7˚N where 24-hour sunlight precluded the identification of twilights in the light intensity data. For geolocators that were not calibrated using either method, we used the calibration parameters from a geolocator that was of the same model and manufactured in the same year (e.g., [58]). This calibration step was necessary to calculate tag-specific parameters that described the difference between the theoretical and observed light-levels recorded by each geolocator [59, 60].

Twilight periods and calibration parameters were then used in the state-space hidden Markov model in FlightR to generate twice-daily location estimates. This approach used observed light-levels, an uncorrelated random walk movement model with migratory and sedentary behavioral states, and a spatially explicit behavioral mask to generate the most probable location estimates according to available information on Dunlin migration ecology [59, 60]. Because Dunlin cannot rest or forage in deep water, we parameterized the movement model so that location estimates over land or over ocean were equally likely if ambient light levels indicated an individual was in a migratory state, but were weighted toward land if light levels indicated an individual was in a sedentary state. Parameter values for the movement model and the spatial behavioral mask were the same for all individuals. We used the "on-the-fly" outlier detection algorithm to discard unrealistic location estimates [59]. In total, 10% of possible location estimates were typically discarded per complete migration track (median: 10%, IQR: 8–11%, $n$ = 84). Finally, we used the stationary.migration.summary function to aggregate daily location estimates and estimate where and when a bird was stationary for 2 days or longer (hereafter, stationary estimates; [60]). These stationary estimates include stopover and staging sites [61], and were used in all further analyses. We chose to use FlightR because daily location estimates, and derived stationary estimates, are less affected by tag shading than a traditional threshold analysis [59], and it provides more reliable estimates in the weeks surrounding the vernal and autumnal equinox [62].

After generating an initial migration track, we re-calibrated each geolocator using the median latitude, longitude, and light level data from the bird's longest estimated stationary site (typically during the boreal winter). We chose to re-calibrate because calibration parameters from a nonbreeding site are better at accounting for environmentally and behaviorally induced noise in light-level recordings at nonbreeding sites than calibration parameters from a rooftop calibration, or while a bird was on its breeding grounds [56, 57]. This re-calibration approach was necessary because the truncated light-level recordings collected by Mk12 tags precluded our ability to use other unknown-location-based calibration methods [e.g., 60, 63]. After re-calibrating, we used the re-calibration parameters to re-estimate the migration track and stationary estimates. Re-estimated migration tracks were typically more precise, having less variability between consecutive location estimates and typically half as many daily location estimates discarded by the outlier detection algorithm (median: 5%, IQR: 4–6%, $n$ = 84). Other than increased track precision, re-estimated migration tracks exhibited no major changes in spatial or temporal itineraries of individual birds (S1 Appendix).

## Refining migration tracks

We further refined stationary estimates and subsequent migration tracks using a 3-step process. First, because of the general inaccuracy of solar geolocation [62, 64, 65], we merged all sequential stationary estimates that were < 250 km apart by averaging geographic coordinates

and combining arrival/departure dates. We also discarded stationary estimates prior to a bird travelling > 250 km from their known breeding site (i.e., capture/re-capture site). This distance is a conservative buffer for the geographic resolution of stationary estimates returned by FlightR [62] and functions to aggregate routine movements that may occur during a stationary period (e.g., daily movements between roosting and foraging sites; [66]. Second, because solar geolocation performs poorly at high latitudes [67], we discarded stationary estimates prior to a bird travelling south of 66.7˚N. Finally, because FlightR can generate erratic stationary estimates due to noisy light-level data [59], we discarded stationary estimates that had a turning angle of < 60˚ (i.e., locations comprising an angle of < 60˚ between their prior and subsequent stationary estimate; [68] but combined their arrival/departure date with their nearest neighbor. We did not follow this procedure for the stationary estimate that was farthest from the breeding site and had a stationary period ≥ 42 days (S1 Appendix). This approach assumes that an individual migrated without reversing direction until they departed their most distant winter site to migrate north to breed [e.g., 69]. It also assumes that an individual would stop at its farthest winter site for ≥ 42 days, a minimum winter duration supported by prior Dunlin tracking studies [58, 70] and repeat band resightings of Dunlin on the EAAF [71]. Although these assumptions might not be fully met (e.g., a bird could exhibit north-south movements during migration or winter), spatial inaccuracies in estimating latitude preclude finer resolving of the tracks.

## Defining migration parameters

To describe the migration ecology of individual Dunlin, we divided each migration track into 4 periods: breeding, south migration, winter, and north migration. We then estimated the following parameters: migration initiation date and arrival date on breeding and wintering grounds, migration duration, minimum migration distance, migration speed (km/day), total number of stationary estimates, and days spent at each stationary estimate. If a bird's first or last stationary estimate was > 250 km from their breeding site (e.g., due to breeding north of 66.7˚N), their south migration initiation/north migration arrival date was back/forward calculated by dividing the minimum geographic distance between their first/last stationary estimate and their breeding site by an assumed travel rate of 58 kilometers per hour [72]. This analysis was unable to consider post- and pre-breeding stationary periods that occurred north of 66.7˚N or were within 250 km of a bird's breeding site and, therefore, southward initiation and northward arrival dates are the latest and earliest dates possible, respectively.

South migration began when an individual departed for a stationary estimate that was south of 66.7˚N and > 250 km from its breeding site. South migration ended, and the winter period began, when an individual arrived at a stationary estimate that was south of 45.5˚N and that lasted for ≥ 42 days. We chose 45.5˚N and ≥ 42 days as the migration-winter threshold because Dunlin on the EAAF overwinter at latitudes this far north [40], are generally stationary through the winter [58, 70, 71], and typically stop at migration sites for < 42 days [58, 70, 73]. The winter period ended, and north migration began, when an individual departed for a stationary estimate north of 45.5˚N (i.e., outside of the typical wintering range), or to a stationary estimate(s) south of 45.5˚N and for < 42 days before migrating north of 45.5˚N. North migration ended when an individual arrived at a stationary estimate that was < 250 km from its breeding site or, if a bird's last stationary estimate was > 250 km from their breeding site (e.g., due to breeding north of 66.7˚N), was forward calculated by estimating the time it would take to travel between its final stationary estimate and its Arctic breeding site (see above). Finally, we defined minimum migration distance as the sum of geographic distances between sequential stationary estimates within south migration, winter, and north migration periods.

We did not estimate the distance of the actual route that the bird flew. We estimated migration speed by dividing the minimum migration distance by the total days spent in each period (i.e., migration duration). Our approach to estimating migration speed included stationary fueling periods, and therefore, slower migration speeds by individuals that migrated farther, and/or crossed major ecological barriers (e.g., the Bering Sea), may reflect a non-linear increase in fueling demands [74] and not necessarily individual differences in migration strategy [75]. To avoid potentially misinterpreting differences in migration speed between individuals, we only assessed within-individual changes in migration speed between south and north migration.

### Evaluating subspecific migration dynamics

Annual migration tracks were combined across individuals to construct a migratory network map that was used to evaluate subspecies' spatial and temporal migration dynamics. Partial migration tracks, resulting from geolocators malfunctioning, were included in the characterization of subspecies' migration ecology (e.g., migration initiation and arrival dates, etc.), but were not included in the migratory network. If a bird was tracked over multiple years, we only used the first year of data.

To construct the migratory network, we first clustered stationary estimates into flyway regions using the partitioning around medoids (pam) function in the raster package (v2.5–8; [76]) in program R. This function required that the number of clusters (i.e., flyway regions) be decided a priori. We optimized the number of clusters by iteratively clustering stationary estimates until the median cluster diameter was < 700 km. We selected a 700 km threshold because resightings of Dunlins carrying geolocators indicated that stationary estimates were typically accurate to within 700 km of an individual's actual location ($n$ = 5; location error = 45 km, 260 km, 349 km, 1067 km, 1246 km). This approach allowed us to objectively define flyway regions at a fine enough spatial resolution to be biologically relevant but coarse enough to capture much of the geographic uncertainty in stationary estimates [62, 64, 65]. We then determined the proportion of each subspecies that occurred in each flyway region on each ordinal day (i.e., day 1–365, independent of year) during south migration, north migration, and winter periods.

Finally, we evaluated the spatial and temporal migration dynamics of Dunlin subspecies using multiple approaches. First, we used pairwise two-tailed Fisher's exact tests to statistically compare the proportion of each subspecies that occurred in each flyway region during south migration, north migration, and winter. We limited these analyses to subspecies with data from ≥ 25 individuals (i.e., *C. a. arcticola* and *sakhalina*). Second, we used pairwise Wilcoxon rank sum tests to evaluate seasonal differences between subspecies in migration initiation date, arrival date, migration duration, minimum migration distance, migration speed (km/day), total number of stationary estimates, and days spent at each stationary estimate. Here, we included all subspecies with ≥ 5 individuals (i.e., *C. a. arcticola*, *sakhalina*, and *kistchinski*).

## Results

In total, 339 geolocators were deployed at 8 sites from 2010 to 2018 (Table 1 and Fig 1). The number of tags deployed ranged from 12 to 145 per site, and 18 to 215 per Dunlin subspecies. One-hundred and seven (32%) tags were recovered, including 84 that recorded complete migration tracks and were used to construct a migratory network, 16 that recorded partial migration tracks and were combined with the above 84 to characterize subspecies' migration parameters (Table 1), 4 that were excluded because they were from a previously tracked bird, and 3 that failed to collect useable data.

## Subspecific migration dynamics

We found that the timing of south migration differed across the subspecies, with *C. a. kistchinski* (*n* = 5) initiating south migration 35–51 days earlier than *C. a. sakhalina* (*n* = 35) and 48–68 days earlier than *C. a. arcticola* (*n* = 59; Wilcoxon rank-sum tests: 95% CI reported above; Table 2). Similarly, *C. a. sakhalina* initiated south migration 10–20 days earlier than *C. a. arcticola* (Table 2). This staggered migration pattern is consistent with our prediction that Dunlin breeding at lower latitudes would initiate south migration earlier than those breeding at higher latitudes (Table 2 and Fig 1). This pattern was violated, however, by the 1 tracked *C. a. actites*, which bred farthest south but migrated later than the other subspecies (Table 2). Subspecies' south migration distances and durations were also similarly staggered (Table 2). The 1 tracked *C. a. actites* had the shortest migration distance and duration (Table 2).

We identified 12 flyway regions that were used only during migration (i.e., migration regions); 2 of the 12 (regions 24 and 25 in W Alaska; Fig 2) were used only during south migration. Regions in W Alaska (regions 24 and 25) and the NW Sea of Okhotsk (region 18) supported a greater number of *C. a. arcticola* (27–56%; *n* = 52) during south migration than *C. a. sakhalina* (0–12%; *n* = 26; pairwise Fisher's exact tests: p = < 0.01–0.01), whereas the NE Sea of Okhotsk (regions 19 and 20) supported a greater number of *C. a. sakhalina* (42% & 69%) than *C. a. arcticola* (10% & 10%, respectively; p = < 0.01; Fig 2). In contrast, the majority of tagged *C. a. arcticola* (56%), *sakhalina* (77%), and *kistchinski* (100%) occurred in N Sakhalin Island (region 15; Fig 2). This is also the only region where *C. a. actites* are known to breed (Fig 1). Although, during south migration *C. a. arcticola*, *kistchinski*, and *sakhalina* occurred in many of the same regions (Fig 2), they generally did not occur in the same region at the same time (Fig 3). Differences in subspecies' south migration initiation dates and durations (see above) were reflected in the subspecies having staggered peak passage dates across migration regions (e.g., region 15, Fig 3), and were consistent with staggered arrival dates on the wintering grounds (Table 2).

During the winter period, Dunlin were generally stationary (Table 3), but *C. a. kistchinski* (*n* = 5) spent 41–88 more days on its wintering grounds than *C. a. sakhalina* (*n* = 26), and 82–112 more days than *C. a. arcticola* (*n* = 52; Wilcoxon rank-sum tests: 95% CI reported above; Table 3). Similarly, *C. a. sakhalina* spent 26–46 more days on its wintering grounds than *C. a. arcticola* (Table 3). However, we found that *C. a. arcticola* were typically more stationary than *C. a. sakhalina*; flying 528–1,455 fewer km, having 1–2 fewer stationary estimates, and spending 14–67 more days at each stationary estimate (Table 3).

**Table 2. South migration characteristics for each subspecies of Dunlin on the East Asian-Australasian Flyway.** Reported is the median value and interquartile range.

| Subspecies | *arcticola* | *sakhalina* | *kistchinski* | *actites* | Wilcoxon rank-sum pairwise comparisons[a] | | |
|---|---|---|---|---|---|---|---|
| **South migration** | ***n* = 59** | ***n* = 35** | ***n* = 5** | ***n* = 1** | **a** | **b** | **c** |
| Initiation | 31 Aug (22 Aug–9 Sep) | 15 Aug (12 Aug–23 Aug) | 5 Jul (29 Jun–9 Jul) | 4 Sep | < 0.01 | < 0.01 | < 0.01 |
| Winter arrival | 2 Nov (28 Oct–13 Nov) | 18 Sep (10 Sep–2 Oct) | 19 Jul (16 Jul–3 Aug) | 8 Sep | < 0.01 | < 0.01 | < 0.01 |
| Total duration (days) | 65 (51–76) | 32 (27–47) | 20 (15–21) | 4 | < 0.01 | < 0.01 | 0.01 |
| Distance (km) | 7,067 (6,433–7,777) | 5,038 (4,786–5,684) | 4,164 (4,110–4,420) | 2,359 | < 0.01 | < 0.01 | 0.01 |
| Number of stationary estimates | 4 (3–5) | 3 (3–5) | 1 (1–2) | 0 | 0.14 | < 0.01 | < 0.01 |
| Stationary estimate duration (days) | 8 (4–17) | 7 (5–11) | 9 (6–13) | – | 0.16 | 0.98 | 0.60 |
| Speed (km/day) | 109 (97–137) | 155 (115–185) | 250 (196–278) | 590 | – | – | – |

[a]"a" indicates a pair-wise comparison between *arcticola* and *sakhalina*; "b": *arcticola* and *kistchinski*; "c": *sakhalina* and *kistchinski*. No statistical comparisons were made with the *actites* subspecies due to a low sample size. P-values are reported.

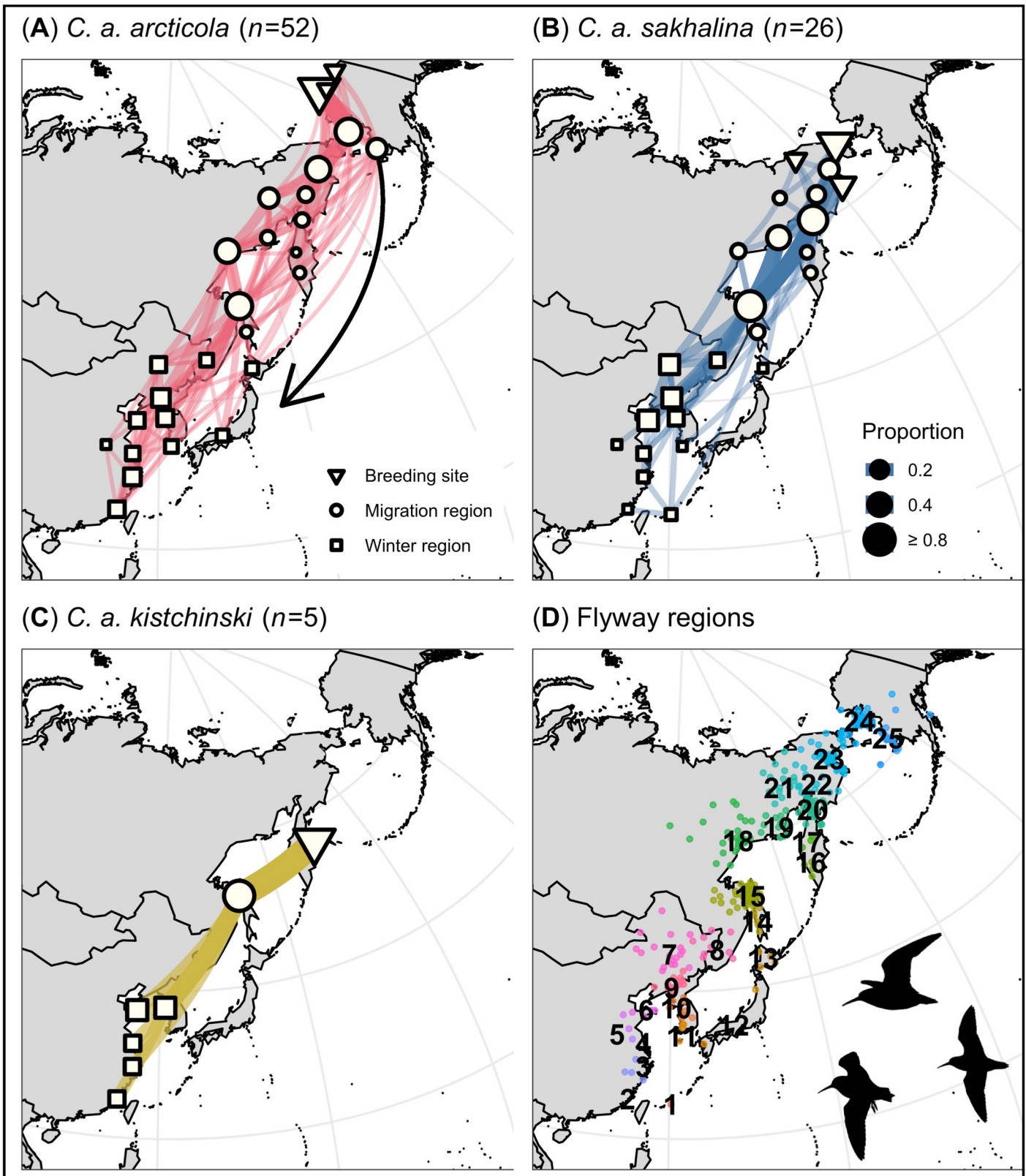

**Fig 2. Migratory network depicting south migration movements made by Dunlin subspecies along the East Asian–Australasian Flyway, and (D) south migration stationary estimates color-coded by flyway region.** Network nodes and edges are weighted by the proportion of individuals that stopped in each flyway region, and the proportion that migrated between flyway regions, respectively.

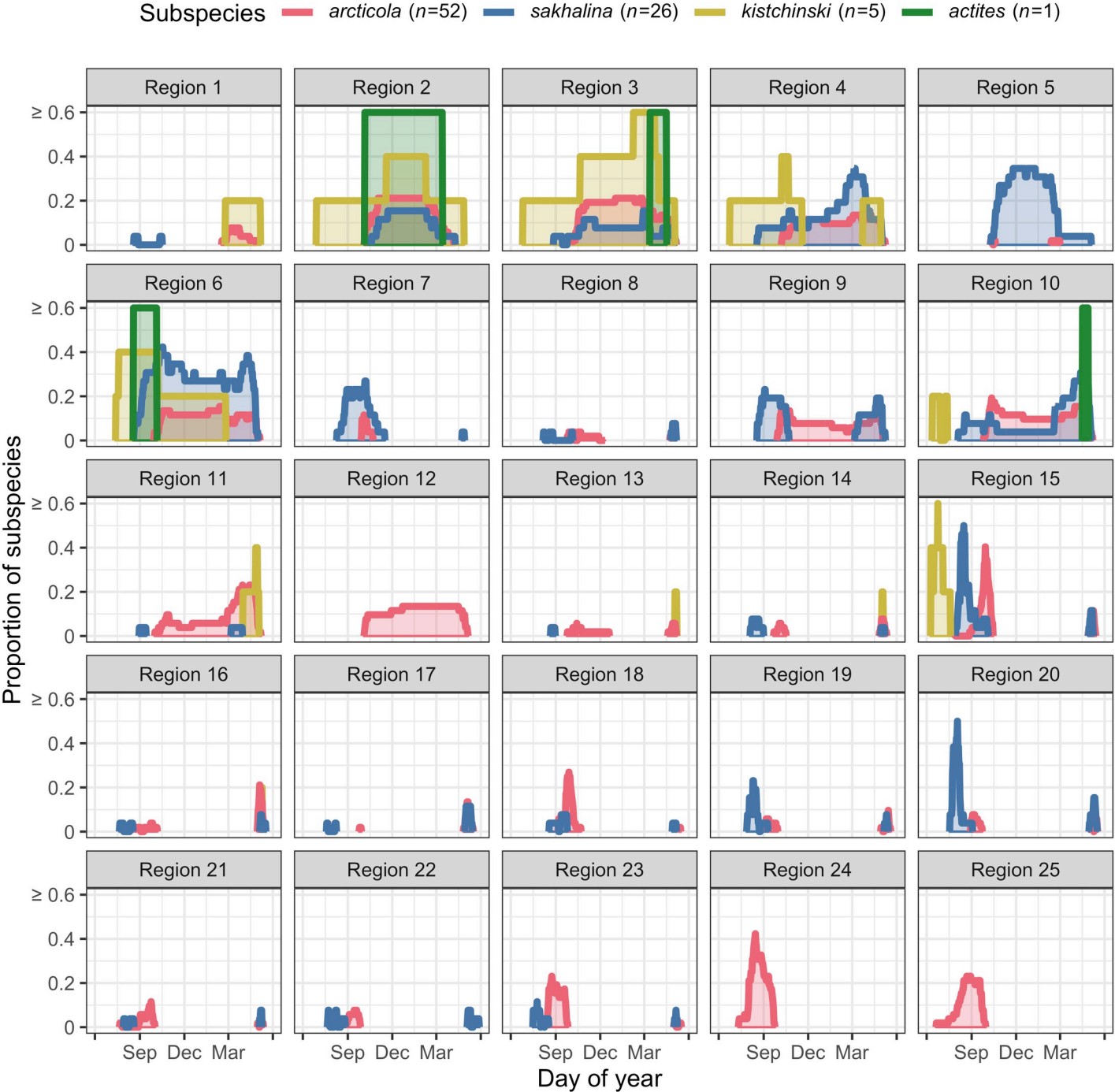

**Fig 3. The proportion of each Dunlin subspecies that occurred in each flyway region along the East Asian-Australasian Flyway by day of year.** See Fig 2 for the location of each region.

We identified 13 flyway regions that were used during the winter (i.e., winter regions). We also found that the S Korean Peninsula and central Japan (region 11 and 12, respectively) supported more *C. a. arcticola* (15% & 14%) than *C. a. sakhalina* (0% & 0%; pairwise Fisher's exact tests: p = 0.05 & 0.09, respectively), whereas inland China, the NW Yellow Sea, and N China (regions 5–7, respectively) supported more *C. a. sakhalina* (19–50%) than *C. a. arcticola* (0–19%;

**Table 3. Winter characteristics for each subspecies of Dunlin on the East Asian-Australasian Flyway.** Reported is the median value and interquartile range.

| Subspecies | *arcticola* | *sakhalina* | *kistchinski* | *actites* | Wilcoxon rank-sum pairwise comparisons[a] | | |
|---|---|---|---|---|---|---|---|
| **Winter** | **n = 52** | **n = 26** | **n = 5** | **n = 1** | **a** | **b** | **c** |
| Total duration (days) | 192 (179–200) | 230 (202–242) | 289 (277–292) | 206 | < 0.01 | < 0.01 | < 0.01 |
| Distance (km) | 0 (0–721) | 1,403 (536–1,953) | 349 (0–1,321) | 1,744 | < 0.01 | 0.29 | 0.29 |
| Number of stationary estimates | 1 (1–2) | 3 (2–3) | 2 (1–2) | 2 | < 0.01 | 0.36 | 0.28 |
| Stationary estimate duration (days) | 128 (56–182) | 66 (49–103) | 102 (81–202) | 47 & 158 | < 0.01 | 0.57 | 0.02 |
| Speed (km/day) | 0 (0–4) | 7 (2–9) | 1 (0–5) | 8 | – | – | – |

[a]"a" indicates a pair-wise comparison between *arcticola* and *sakhalina*; "b": *arcticola* and *kistchinski*; "c": *sakhalina* and *kistchinski*. No statistical comparisons were made with the *actites* subspecies due to a low sample size. P-values are reported.

p = < 0.01–0.01; Fig 4). However, many winter regions supported similar proportions of the subspecies (Fig 4). For example, when assessing major geographic regions, we found that the Yellow Sea (regions 4, 6, 9–11) supported the majority of tagged *C. a. kistchinski* (60%), *arcticola* (63%), and *sakhalina* (92%; including the 1 *C. a. actites*). The East and South China seas (regions 1–3) also supported many *C. a. sakhalina* (31%), *arcticola* (48%), and *kistchinski* (100%; including the 1 *C. a. actites*; Fig 4 and S1 Dataset). No individual Dunlin that wintered in Japan also wintered in the Yellow Sea, China seas, or inland China, and vice-versa (Fig 4). Where subspecies used the same winter region, they generally occurred in the region at the same time (Fig 3).

We found that north migration parameters were similar across the Dunlin subspecies. For example, *C. a. kistchinski*, *sakhalina*, and *arcticola* had similar north migration initiation dates and durations (Table 4). However, like south migration, subspecies migrated different distances, consistent with the minimum distances required to travel between winter regions and subspecific breeding sites (Tables 2 and 4). We also found that individuals differed in how they migrated during north versus south migration. Individuals migrating north typically traveled 1.7–5.4x faster, had 0.2–0.7x as many stationary estimates, and spent 0.3–1.1x the number of days at each stationary estimate than when they migrated south (IQR reported above, *n* = 83; Tables 2 and 4). However, the 1 tracked *C. a. actites* showed the opposite pattern during north migration, migrating slower and stopping more often and for longer durations than during south migration (Tables 2 and 4).

**Table 4. North migration characteristics for each subspecies of Dunlin on the East Asian-Australasian Flyway.** Reported is the median value and interquartile range.

| Subspecies | *arcticola* | *sakhalina* | *kistchinski* | *actites* | Wilcoxon rank-sum pairwise comparisons[a] | | |
|---|---|---|---|---|---|---|---|
| **North migration** | **n = 52** | **n = 26** | **n = 5** | **n = 1** | **A** | **b** | **c** |
| Initiation | 17 May (7 May–19 May) | 17 May (29 Apr–21 May) | 16 May(19 Apr–22 May) | 2 Apr | 0.84 | 1.00 | 0.87 |
| Breeding arrival | 30 May (27 May–3 Jun) | 30 May (27 May–2 Jun) | 27 May (25 May–29 May) | 18 May | 0.53 | 0.05 | 0.10 |
| Total duration (days) | 13 (10–25) | 13(10–31) | 9 (5–40) | 46 | 0.83 | 0.58 | 0.61 |
| Distance (km) | 6,308 (5,806–6,897) | 5,152 (4,944–5,616) | 4,612 (4,562–4,705) | 4,254 | < 0.01 | < 0.01 | < 0.01 |
| Number of stationary estimates | 2 (1–2) | 2 (1–3) | 1 (1–2) | 2 | 0.70 | 0.39 | 0.34 |
| Stationary estimate duration (days) | 5 (3–9) | 5 (3–10) | 3 (3–25) | 11 & 33 | 0.96 | 0.53 | 0.47 |
| Speed (km/day) | 432 (270–615) | 400 (172–494) | 542 (112–922) | 92 | – | – | – |

[a]"a" indicates a pair-wise comparison between *arcticola* and *sakhalina*; "b": *arcticola* and *kistchinski*; "c": *sakhalina* and *kistchinski*. No statistical comparisons were made with the *actites* subspecies due to a low sample size. P-values are reported.

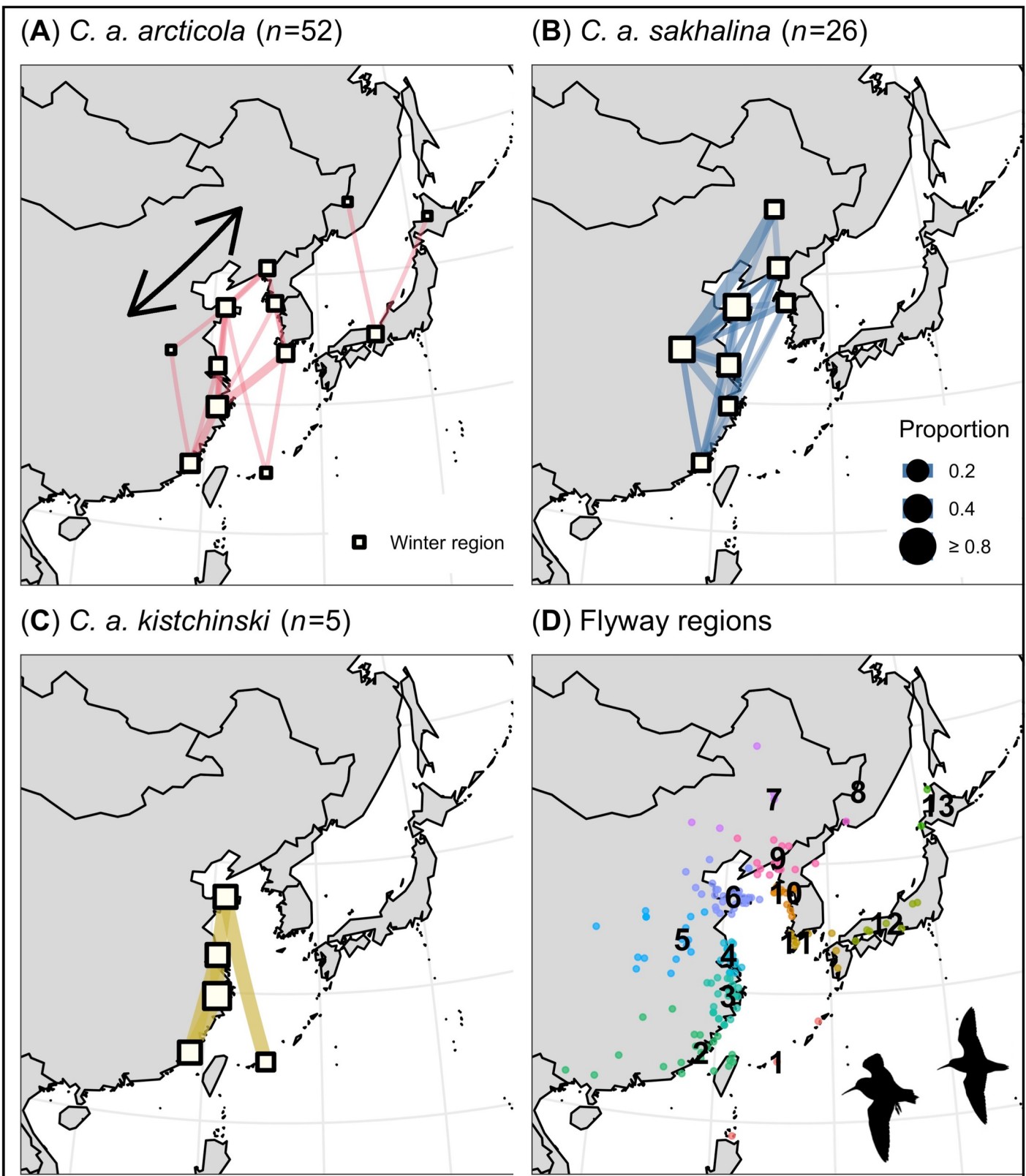

**Fig 4. Migratory network depicting winter movements made by Dunlin subspecies along the East Asian–Australasian Flyway, and (D) winter stationary estimates color-coded by flyway region.** Network nodes and edges are weighted by the proportion of individuals that stopped in each flyway region, and the proportion that migrated between flyway regions, respectively.

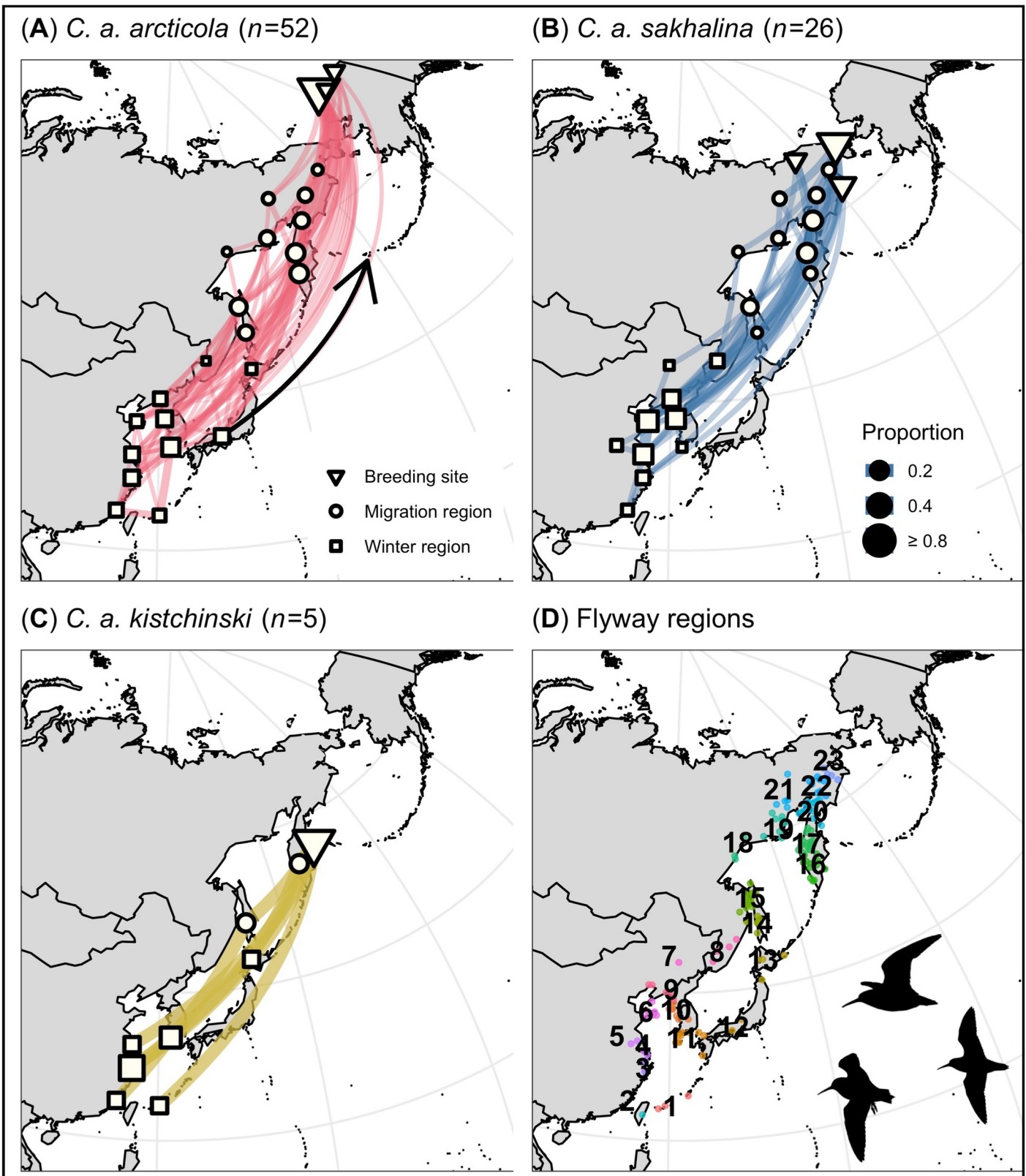

**Fig 5. Migratory network depicting north migration movements made by Dunlin subspecies along the East Asian–Australasian Flyway, and (D) north migration stationary estimates color-coded by flyway region.** Network nodes and edges are weighted by the proportion of individuals that stopped in each flyway region, and the proportion that migrated between flyway regions, respectively.

We identified 10 migration regions that were used during north migration. Across the 10 regions, *C. a. arcticola* and *sakhalina* occurred in similar proportions (Fig 5), had similar peak passage dates (e.g., region 15; Fig 3), and had similar arrival dates on their breeding grounds (Table 4). Over the entire nonbreeding period, Dunlin subspecies exhibited the highest degree of spatial and temporal overlap during north migration, a moderate degree during the winter, and the lowest degree during south migration (Tables 2–4 and Figs 2–5).

## Discussion

Patterns of spatial and temporal flyway use can have a profound effect on individual fitness, and thereby, where and when a population experiences decline [13–15]. For example, the timing and degree to which Dunlin subspecies use an area may affect individuals' access to optimal foraging conditions [77], exposure to predation pressure [78, 79], subsequent reproductive success [1, 77], and survival rates [1]. Our migratory network provided an informative framework for objectively delineating flyway regions and describing population-specific migration patterns in space and time. We found that Dunlin subspecies used many of the same core flyway regions (e.g., the East China Sea, Yellow Sea, and N Sakhalin Island; Figs 2–5), but that *C. a. arcticola* and *sakhalina* segregated along edge flyway regions; with *C. a. arcticola* occurring more along the eastern edge of the flyway (e.g., migrating and wintering in western Alaska and central Japan) and *C. a. sakhalina* occurring more along the western edge of the flyway (e.g., wintering in N China and inland China; Figs 2 and 4). No individual Dunlin that wintered in Japan also wintered in the Yellow Sea, China seas, or inland China, and vice-versa (Fig 4). This apparent east-west divide, combined with evidence that *C. a. arcticola* exhibit strong interannual site fidelity to specific wintering sites in Japan and elsewhere [40], suggests that wintering-ground effects on population survival rates likely operate independently among *C. a. arcticola* that winter in Japan, and among Dunlin that winter elsewhere. Future efforts to compare seasonal survival rates of *C. a. arcticola* that winter in Japan to those of Dunlin that winter outside of Japan could enable researchers to better parse the many factors potentially driving *C. a. arcticola* population declines [26, 27, 80, 81].

During south migration, we found that *C. a. arcticola*, *sakhalina*, and *kistchinski* staggered their migration initiation and winter arrival dates; the southernmost (*kistchinski*) and northernmost (*arcticola*) breeding subspecies migrating first and last, respectively (Table 2 and Fig 1). Although we expected this pattern to result from earlier breeding phenology at lower latitudes [44, 45], it may also reflect differing strategies for how subspecies minimize their exposure to predation by Peregrine Falcons (*Falco peregrinus*); *C. a. kistchinski* timed their south migration before Peregrine Falcons typically migrate south [82], and *C. a. arcticola* migrated after (e.g., that seen in Western Sandpiper [*Calidris mauri*] and Dunlin on the East Pacific Flyway, [78]; Table 2). Alternatively, subspecific differences in south migration timing may reflect differing strategies for where and when subspecies undergo flight feather molt; *C. a. arcticola* and *sakhalina* undergo flight feather molt on their breeding grounds (mid-June to late August; [45, 83], whereas, *C. a. kistchinski* initiate flight feather molt in mid-July (on their breeding [45] or nonbreeding grounds [84] and complete it on China Sea and Yellow Sea nonbreeding grounds (e.g., that seen in Dunlin in the western Palearctic, [85]; Table 2, [84]. If the south migration initiation dates we observed are widespread among *C. a. kistchinski*, efforts to identify and conserve important China Sea and Yellow Sea molting sites may be more critical to the persistence of this subspecies than previously recognized [33, 86].

During the winter, we found that *C. a. sakhalina* flew farther, stopped more, and spent fewer days at each stop than *C. a. arcticola* (Table 3). The more mobile wintering behavior exhibited by *C. a. sakhalina* likely reflects the subspecies unique use of freshwater wetlands in

inland China (region 5; Fig 4) and annual hydrological dynamics in the Yangtze River Floodplain (YRF), where most Dunlin in inland China occur [87]. Indeed, we found that 46% of tagged *C. a. sakhalina* wintered in inland China. All arrived after monsoonal flood waters typically recede and reveal abundant shorebird habitats (i.e., October), and all generally departed before the spring rains (i.e., April; Fig 3, [88, 89]); requiring individuals make additional migrations between inland China and alternate wintering regions in the fall and spring (e.g., that seen in *C. a. pacifica* on the East Pacific Flyway, [90]; S1 Dataset). In addition to flying farther and stopping more, *C. a. sakhalina* likely comprises much of the > 45,000 Dunlin that occur in inland China [87], and is likely the subspecies most threatened by habitat degradation/loss from human modifications to the natural hydrological regime of the YRF (e.g., the Three Gorges Dam; [88, 89, 91]. However, many waterbodies in the YRF are connected through a series of sluices, and therefore, coordination of hydrological management actions that optimize wetland habitat quality and seasonal availability could significantly support *C. a. sakhalina* populations in the region [92].

During north migration, we found that *C. a. arcticola*, *sakhalina*, and *kistchinski* departed their wintering sites and arrived at their breeding sites on similar dates (Table 4), despite differences in their migration distances (Table 4) and breeding phenologies [44, 45]. However, arrival dates for *C. a. arcticola* and *sakhalina* that bred north of 66.7˚N may have been 3–10 days later than estimated [93] because we were unable to identify pre-breeding stationary periods in areas with 24-hour sunlight (see above). Nonetheless, the similar departure dates suggest that Dunlin subspecies use similar social and/or environmental cues, such as annual changes in day length [94], to time their north migrations.

## Limitations and future directions

The migration patterns presented here are for adult Dunlin that returned to the same breeding site and were recaptured in a following year. Therefore, our results do not include the migration patterns of birds that died, juveniles, or adults that emigrated from their original capture site. Uneven sampling effort across subspecies and breeding sites is another limitation of our findings. For example, the migration patterns we found for *C. a. actites* and *kistchinski* are from 1 and 5 individuals, respectively, and each subspecies was only studied at a single breeding site (Table 1). Lastly, the geographic uncertainty associated with geolocator-derived location estimates [62, 64, 65] required coarse spatial and temporal interpretation [56].

Combining migration tracking data with on-the-ground survey techniques may be an effective approach to refine our understanding of subspecies' migration patterns and to overcome the limitations of our findings. For example, *C. a. actites* is classified as vulnerable under the International Union for Conservation of Nature's regional Red List criteria, due to its small population size [39]. Despite conservation concerns, targeted conservation efforts have not been possible because, until recently, little was known of the subspecies' migration dynamics [39, 40]. Our tracking data and previously published band recoveries [40, 95] indicate the subspecies migrates through the Yellow and East China seas and primarily winters in the South China Sea (Fig 3). Due to *C. a. actites*' genetic and morphological distinctness among Dunlin subspecies [96, 97], it is possible to combine capture and sampling methodologies with flock counts [e.g., 43, 98] to estimate how many *C. a. actites* likely occur at particular South China Sea sites, and at sites in other regions where *C. a. actites* occur [40, 95]. Collectively, such efforts may significantly advance our understanding of *C. a. actites* space-time dynamics and our ability to implement conservation actions for this vulnerable subspecies.

Combining migration tracking data with on-the-ground survey techniques may also be an effective approach to estimate subspecies' population sizes and long-term trends. For example,

we found that subspecific migration phenologies were generally asynchronous during south migration, and that the pattern of asynchrony was consistent across migration regions (Fig 3 and S1 Table). By understanding how subspecies migrate in temporally distinct waves, on-the-ground survey efforts (e.g., daily flock counts) may be combined with migration tracking data (e.g., individual turnover rates, population peak passage dates) to estimate the number of individuals of each subspecies' that use a particular site [99, 100]. Such survey efforts may also be applied across years and across key regions to estimate each subspecies' population size [87, 101] and population trends [2, 102, 103]. Such a monitoring design could provide specific information necessary to inform more comprehensive conservation plans for Dunlin on the EAAF [33, 38, 39].

## Conclusion

Our migratory network, constructed using geolocator-derived migration tracks of individual Dunlin, provided an informative framework for objectively delineating flyway regions and describing population-specific migration patterns in space and time. We found Dunlin subspecies exhibited unique patterns of spatial and temporal flyway use on the EAAF. Spatially, *C. a. arcticola* predominated in regions along the eastern edge of the flyway (e.g., western Alaska and central Japan), whereas *C. a. sakhalina* predominated in regions along the western edge of the flyway (e.g., N China and inland China; Figs 2 and 4). No individual Dunlin that wintered in Japan also wintered in the Yellow Sea, China seas, or inland China, and vice-versa (Fig 4). However, similar proportions of the 4 subspecies used many of the same regions at the center of the flyway (e.g., N Sakhalin Island and the Yellow Sea; Figs 2–5). Temporally, Dunlin subspecies staggered their south migrations and exhibited little temporal overlap among subspecies within shared migration regions (Table 2 and Fig 3). In contrast, Dunlin subspecies migrated simultaneously during north migration (Table 4 and Fig 3). South migration was also characterized by individuals stopping more often and taking more days to complete their migration (Table 2) than during north migration (Table 4). Taken together, these spatial-temporal migration dynamics indicate that Dunlin subspecies may be differentially affected by regional habitat change and population declines according to where and when they occur. By understanding how subspecies migrate south in temporally distinct waves (S1 Table), we suggest on-the-ground survey efforts (e.g., daily flock counts) may be combined with migration tracking data (e.g., individual turnover rates, population peak passage dates) to estimate the number of individuals of each subspecies' that use a particular site [99, 100]. Such survey efforts may also be applied across years and across key regions to estimate subspecies' population sizes [87, 101] and long-term trends [2, 102, 103]. Such studies would significantly advance our understanding of Dunlin space-time dynamics and the coordination of Dunlin conservation actions across the EAAF.

## Supporting information

**S1 Appendix. Light-level geolocator analyses.** Annotated R code of steps taken to generate geolocator-derived stationary estimates and refine Dunlin migration tracks along the East Asian-Australasian Flyway.
(HTML)

**S1 Dataset. Light-level geolocator location data.** Geolocator-derived stationary estimates comprising 100 Dunlin migration tracks along the East Asian-Australasian Flyway. A ReadMe tab is included to help guide the user.
(XLS)

**S1 Table. South migration timing of Dunlin subspecies by migration region.**
(PDF)

**S1 Fig. South migration characteristics for each subspecies of Dunlin on the East Asian-Australasian Flyway.** Reported is the median value and interquartile range.
(TIF)

**S2 Fig. Winter characteristics for each subspecies of Dunlin on the East Asian-Australasian Flyway.** Reported is the median value and interquartile range.
(TIF)

**S3 Fig. North migration characteristics for each subspecies of Dunlin on the East Asian-Australasian Flyway.** Reported is the median value and interquartile range.
(TIF)

## Acknowledgments

We thank the many people that lent their expertise in marking and recapturing the Dunlin in this study, including Julia Bojarinova, Jenny Cunningham, Andy Doll, John Diener, Scott Freeman, Anna Gleizer, Lizzie Goodrick, Kirsten Grond, Brooke Hill, Sarah Hoepfner, Denis Irinjakov, Lindall Kidd, Alan Kneidel, McKenzie Mudge, Kevin Piertzak, Ron Porter, Martin Robards, Shiloh Schulte, Tatiana Svatko, Natalia Vartanyan, Stephen Yezerinac, and Steve Zack. We also thank the people of Utqiaġvik, Alaska and the Ukpeaġvik Iñupiat Corporation for allowing us to conduct work on their lands. Finally, we appreciate the comments provided by Chi-Yeung Choi, Simeon Lisovski, an anonymous reviewer, and our peers at the University of Colorado Denver. They greatly improved this manuscript.

## Author Contributions

**Conceptualization:** Benjamin J. Lagassé, Richard B. Lanctot, Michael B. Wunder.

**Data curation:** Benjamin J. Lagassé, Rebecca L. McGuire.

**Formal analysis:** Benjamin J. Lagassé.

**Funding acquisition:** Benjamin J. Lagassé, Richard B. Lanctot, Stephen Brown, Steve Kendall, Christopher J. Latty, Joseph R. Liebezeit, Egor Y. Loktionov, Rebecca L. McGuire, David C. Payer, Diana V. Solovyeva, Pavel S. Tomkovich, Olga P. Valchuk.

**Investigation:** Benjamin J. Lagassé, Richard B. Lanctot, Alexei G. Dondua, Joseph R. Liebezeit, Egor Y. Loktionov, Konstantin S. Maslovsky, Alexander I. Matsyna, Ekaterina L. Matsyna, Diana V. Solovyeva, Pavel S. Tomkovich, Olga P. Valchuk.

**Methodology:** Benjamin J. Lagassé, Richard B. Lanctot, Michael B. Wunder.

**Project administration:** Benjamin J. Lagassé, Richard B. Lanctot, Stephen Brown, Alexei G. Dondua, Steve Kendall, Christopher J. Latty, Joseph R. Liebezeit, Egor Y. Loktionov, Konstantin S. Maslovsky, Alexander I. Matsyna, Ekaterina L. Matsyna, Rebecca L. McGuire, David C. Payer, Sarah T. Saalfeld, Jonathan C. Slaght, Diana V. Solovyeva, Pavel S. Tomkovich, Olga P. Valchuk.

**Supervision:** Richard B. Lanctot, Michael B. Wunder.

**Validation:** Benjamin J. Lagassé.

**Visualization:** Benjamin J. Lagassé.

Writing – **original draft:** Benjamin J. Lagassé.

Writing – **review & editing:** Benjamin J. Lagassé, Richard B. Lanctot, Stephen Brown, Alexei G. Dondua, Steve Kendall, Christopher J. Latty, Joseph R. Liebezeit, Egor Y. Loktionov, Konstantin S. Maslovsky, Alexander I. Matsyna, Ekaterina L. Matsyna, Rebecca L. McGuire, David C. Payer, Sarah T. Saalfeld, Jonathan C. Slaght, Diana V. Solovyeva, Pavel S. Tomkovich, Olga P. Valchuk, Michael B. Wunder.

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
