## [Decision Letter · Decision Letter 0]

2 May 2022

PONE-D-22-05779Flyway network model reveals unique spatial-temporal migration dynamics of Dunlin subspecies along the East Asian-Australasian FlywayPLOS ONE

Dear Dr. Lagasse,

Thank you for submitting your manuscript to PLOS ONE. After careful consideration, we feel that it has merit but does not fully meet PLOS ONE’s publication criteria as it currently stands. Therefore, we invite you to submit a revised version of the manuscript that addresses the points raised during the review process.

Your revised manuscript can be accepted for publication provided that you address the remaining reviewer concerns. Please respond carefully to the reviewers’ points indicated below, especially for the Reviewer 2.

We look forward to receiving your revised manuscript.

Kind regards,

Peng Chen, Ph.D.

Academic Editor

PLOS ONE

Journal Requirements:

(This work received financial and/or logistical support from the American Ornithological Society–Alexander Wetmore Research Award, NGO Amur–Ussuri Center for Avian Biodiversity (Vladivostok, Russia), Arctic Landscape Conservation Cooperative, Arctic Shorebird Demographics Network, BirdsRussia, British Petroleum Exploration (Alaska) Inc., Bureau of Land Management, Calvin J. Lensink Fund, Kinross Gold Corporation–Kupol Mine, The MacArthur Foundation, Manomet Inc., National Fish and Wildlife Foundation, Neotropical Migratory Bird Conservation Act Grant Program, The Nuttall Ornithological Club–Blake-Nuttall Fund Grant, The Trust for Mutual Understanding, University of Alaska Fairbanks, University of Colorado Denver, University of Missouri Colombia, U.S. Fish and Wildlife Service (Avian Influenza Program, Migratory Bird Management Division, National Wildlife Refuge Challenge Cost Share Program, National Wildlife Refuge Division), Wildlife Conservation Society–Arctic Beringia Regional Program, and the Wilson Ornithological Society–Paul A. Stewart Grant. The funders had no role in study design, data collection and analysis, decision to publish, or preparation of the manuscript.)

(Please include your amended statements within your cover letter; we will change the online submission form on your behalf.

This work received financial and/or logistical support from the American Ornithological Society–Alexander Wetmore Research Award, NGO Amur–Ussuri Center for Avian Biodiversity (Vladivostok, Russia), Arctic Landscape Conservation Cooperative, Arctic Shorebird Demographics Network, BirdsRussia, British Petroleum Exploration (Alaska) Inc., Bureau of Land Management, Calvin J. Lensink Fund, Kinross Gold Corporation–Kupol Mine, The MacArthur Foundation, Manomet Inc., National Fish and Wildlife Foundation, Neotropical Migratory Bird Conservation Act Grant Program, The Nuttall Ornithological Club–Blake-Nuttall Fund Grant, The Trust for Mutual Understanding, University of Alaska Fairbanks, University of Colorado Denver, University of Missouri Colombia, U.S. Fish and Wildlife Service (Avian Influenza Program, Migratory Bird Management Division, National Wildlife Refuge Challenge Cost Share Program, National Wildlife Refuge Division), Wildlife Conservation Society–Arctic Beringia Program, and the Wilson Ornithological Society–Paul A. Stewart Grant. The funders had no role in study design, data collection and analysis, decision to publish, or preparation of the manuscript.)

(This work received financial and/or logistical support from the American Ornithological Society–Alexander Wetmore Research Award, NGO Amur–Ussuri Center for Avian Biodiversity (Vladivostok, Russia), Arctic Landscape Conservation Cooperative, Arctic Shorebird Demographics Network, BirdsRussia, British Petroleum Exploration (Alaska) Inc., Bureau of Land Management, Calvin J. Lensink Fund, Kinross Gold Corporation–Kupol Mine, The MacArthur Foundation, Manomet Inc., National Fish and Wildlife Foundation, Neotropical Migratory Bird Conservation Act Grant Program, The Nuttall Ornithological Club–Blake-Nuttall Fund Grant, The Trust for Mutual Understanding, University of Alaska Fairbanks, University of Colorado Denver, University of Missouri Colombia, U.S. Fish and Wildlife Service (Avian Influenza Program, Migratory Bird Management Division, National Wildlife Refuge Challenge Cost Share Program, National Wildlife Refuge Division), Wildlife Conservation Society–Arctic Beringia Regional Program, and the Wilson Ornithological Society–Paul A. Stewart Grant. The funders had no role in study design, data collection and analysis, decision to publish, or preparation of the manuscript.)

Reviewers' comments:

Reviewer's Responses to Questions

**Comments to the Author**

1. Is the manuscript technically sound, and do the data support the conclusions?

Reviewer #1: Yes

Reviewer #2: Partly

Reviewer #3: Yes

2. Has the statistical analysis been performed appropriately and rigorously? 

Reviewer #1: Yes

Reviewer #2: Yes

Reviewer #3: Yes

3. Have the authors made all data underlying the findings in their manuscript fully available?

Reviewer #1: Yes

Reviewer #2: Yes

Reviewer #3: Yes

4. Is the manuscript presented in an intelligible fashion and written in standard English?

Reviewer #1: Yes

Reviewer #2: Yes

Reviewer #3: Yes

5. Review Comments to the Author

Reviewer #1: Thanks for sending me this interesting manuscript PONE-D-22-05779 by Benjamin Lagasse et al. titled “Flyway network model reveals unique spatial-temporal migration dynamics of Dunlin

subspecies along the East Asian-Australasian Flyway”. I am glad to see the data from this amazing long-term collaborative tracking effort being analyzed and written up. The authors summarized the migration characteristics of four Dunlin subspecies along the flyway. Despite being a common species along the flyway, we know very little about the migration pattern among different subspecies, especially when observing them in the non-breeding grounds. This study also provides important insights in monitoring and conservation effort and I added a few comments below which could be used to improve the manuscript.

General comments:

1. I agree with the authors that this work is useful for guiding on-the-ground survey efforts. It would be even more useful if the authors can be more specific here, either in the main text or as supplementary materials, to state when is the good time for a particular region to conduct its Dunlin survey (probably the time when only one subspecies occur?) to achieve long-term monitoring goals.

2. Line 468 – does this mean no sakhalina occured in the two regions? It's unclear if this percentage represent the subspecies composition within one region, or regional composition of one subspecies. I believe it's the latter, and if that's the case, then the presentation using region as the subject in a sentence to report the percentage is confusing (Sorry if I get this wrong). The same applies to the rest of this paragraph, it would help to clarify this at the beginning of this paragraph.

3. Line 565 – The departure date seems to be different to some published records (https://cms.hkbws.org.hk/cms/component/phocadownload/category/25-reports-of-shorebird-monitoring) and my personal observation in southern China such as Leizhou Peninsula. Dunlin in southern China 'disappear' in March or even earlier. It would be good to mention this phenomenon and the potential reasons behind the difference in those records with this geolocator study. Base on the definitions used in this study and the resolution of geolocator data, could a short move from southern China to the Yellow Sea be captured in the analysis?

4. Lines 633-635 – note that this seems to contradict to what was said Lines 484-487.

Sorry if I misunderstood something here.

Specific comments:

Lines 77: This sentence appears in the manuscript several times but please consider reword it a bit to explain clearly what was meant to say. An individual can 'winter' only in one place isnt'? Perhaps 'No individual Dunlin demonstrated movement between ... within the same winter'?

Lines 104, 109: Consider ‘population decline’

Line 131: instead of 'migration sites', consider 'stopping and wintering sites'

Line 132: Consider citing Zhang et al's work, which showed a different way of degradation than the three references cited (doi:10.1017/S0959270917000430)

Line 140: please also report the rates

Line 142: instead of M. Tian pers. comm, now there is published record for this “a 28.4% decline in the Chinese Yellow Sea within about a decade” (doi.org/10.1016/j.biocon.2022.109547)

Line 171: is there a name for this pattern? Parallel migration?

Line 178: Should reword "which breed between the 2 other subspecies" as this sounds a bit strange.

Lines 533, 534: Perhaps ‘times’ instead of ‘timing’?

Reviewer #2: First, I would like to congratulate the authors presenting a comprehensive analysis of the migration routes and timing and wintering movements of several subspecies of Dunlins in the East Asian-Australasian Flyway (EAAF). This is a key contribution towards conservation of this species in the EAAF.

One thing that really puzzles me is the title and framing of the manuscript as presenting a ‘flyway network model’, since I could not find any network analyses being presented in the manuscript. For a manuscript about a network model, I would expect the calculation of network metrices such as relative node strength and betweenness centrality (see Jacoby and Freeman 2016 for a good overview of methods applicable in movement ecology). In the second paragraph of the Introduction, several papers that applied network theory to evaluate patterns of migratory connectivity, identifying important sites, etc. were cited. Therefore, I am surprised not finding the analyses.

Unless the authors would like to conduct additional analyses based on network theory, I would suggest framing the manuscript differently. The results presented are valuable to understand the migration ecology of this species and highly informative in its conservation. I suggest deleting the ‘flyway network model’ in the title. For example, ‘Migration patterns of four Dunlin subspecies along the East Asian-Australasian Flyway’ would be a more appropriate title. Accordingly, the Introduction would need to be heavily revised, as well as other parts of the manuscript.

Further comments:

Line 72: delete ‘to construct a flyway network model’ and rewrite

Line 104: I don’t get why ‘however’ is used. There are some studies that focuses on spatial connectivity, but there are also a lot of studies about both spatial and temporal aspects of migration.

Line 126: The subtitle ‘study system’ in the Introduction is not necessary.

Line 154: It is good to mention results of Lagasse et al. (2020a), however too many details are included. This paragraph can be shortened substantially and combined with the previous one. For example, you can try to summarise it in one sentence starting with, ‘Recent analysis of band recoveries reveals…’

Line 166: I think there needs to be more elaboration on why knowledge on migration patterns is informative in developing flyway conservation actions. You can introduce the importance of identifying stopover sites here, as this is a main result of the manuscript but not much is said about it in the Introduction.

Line 169: The authors should rethink what is the main aim and objectives of this manuscript and revise this paragraph accordingly.

Line 263: Instead of stating that Dunlin is a ‘terrestrial species’, it is better to specify that Dunlins cannot rest or forage in deep water.

Line 267: ‘same for all Dunlin’: say ‘same for all individuals’ instead.

Line 271: which R package is this ‘stationary.migration.summary’ function from?

Line 273: Please elaborate the biological meaning of ‘stationary estimates’, either here or at another part of the manuscript. This is used a lot in the Results and it will help interpretation of your findings to link it to more common terms, such as a stopover site or staging site.

Line 298-300: I don’t understand this sentence, what exactly are you trading-off?

Line 314: I do not understand this procedure. Why do you discard stationary estimates that had a turning angle of <60 degrees? If you assume an individual migrated without reversing direction (line 319), then you should discard points that have a large turning angle instead? I could not find this procedure described in the reference given - Edelhoff et al. 2016 (line 316). I wonder how discarding these stationary estimates would affect the resulting flyway regions.

Line 363: It is hard to understand the phrase ‘to avoid potentially misinterpreting differences in migration speed between individuals’ – what types of misinterpretation are you avoiding? I think it made sense to look at within-individual changes in speed between north and south migration, maybe just rewrite the reasoning.

Line 368: I do not think the analyses described here is about constructing a flyway network model. It is just a method to cluster the stationary estimates.

Line 375: In this paragraph, you described how ‘flyway region’ is extracted, but ‘flyway region’ is an uncommon term and you should explain what it means in terms of stopover ecology. I imagine it could be a cluster of stopover sites since a ‘normal’ stopover site would have a much smaller diameter.

Line 413: do you know the sex of this C. a. actites individual? Perhaps it is the sex that takes care of the young. The timing differences between the other three populations that you found could also be due to differences in breeding success and the sexes that were tracked. Do you know the sex of the birds and whether some tagged birds had nest failures? Failed breeders might leave for southward migration earlier. I think an analysis that takes these two factors into account is necessary to really say something about differences between the subspecies. Hatching success should be able to be determined from the geolocator data by extracting the incubation period (e.g. Verhoeven et al. 2020).

Fig 2: This is not a flyway network model. It does not describe the relationships between the flyway regions statistically or mathematically (e.g. based on graph theory) other than plotting a line between them. The caption would be fine if you start with ‘General (A) south migration…’.

Also, the part ‘migrating and wintering birds’ in the sentence ‘see methods for determining migration and wintering regions…’ seems redundant.

Among figure 2A to D, I find only figure 2D is informative. The circles/triangles/squares in Fig A to C are quite misleading as the actual stationary estimates that each shape represents covers a much larger area, according to Fig 2D. Instead of these diagrams, I would really like to see the migration tracks used by different subspecies of Dunlin plotted with a different colour to show the differences and overlaps between subspecies’ migration routes (the colours can be slightly transparent to show overlaps). Such a map will illustrate what you described in line 512-519.

Line 497: To be consistent with the wording, should it be ‘flyway regions’ instead of ‘migration regions’?

Line 532: do you have any information on migration phenology of Peregrine Falcons in the EAAF? If not, there is really no support for this guess. Even Lank et al. (2003) have presented alternative hypotheses that are not danger-based. I suggest changing this sentence to reflect that there are several hypotheses (not mutually exclusive) that can explain these patterns.

Line 574-578: The logic of this sentence does not make sense. Your results agree with the general predictions that migrants used a ‘time minimization strategy’ during northward migration and ‘energy minimization strategy’ during southward migration (Alerstam and Lindström 1990). But this strategy itself does not lead to optimal timing of arrival, which is achieved by migrating at the right time, e.g. by using environmental cues that predict breeding ground food availabilities. Also, the strategy does not lead to higher stress levels. The papers you cited at line 577-578 are about species that are facing habitat degradation during northward migration. These negative impacts are not expected in situations with good stopover habitat quality.

Line 581: In this paragraph you should also discuss that the southward migration timing could be a result of breeding success and the sex of the individuals tracked.

Line 612: if the subspecies are not distinguishable from each other during field observations, it seems impossible to estimate, from flock counts and tracking data, the number of individuals of each subsepecies that use a particular site. To achieve that would also require knowing the population size of each subspecies and the proportion of the population that uses a particular stopover site. The latter might be able to be derive from tracking data, but that would require a very large sample size.

Line 614, 639: To estimate population size and trends, the obvious method would be counting at wintering regions. I wonder how that could be done at stopover sites (as suggested in these lines) where several subspecies co-occur and you cannot visually distinguish them from each other. If you have a valid method in mind, please elaborate more.

Line 638: It would be better to specify which part of the results is useful in guiding on-the-ground survey efforts. I think spatially it is not really useful as the area indicated (including the uncertainty of 200 km) is way too large. Only the timing of when Dunlins occur at a specific area would be useful. From the point of view of somebody doing ground surveys, it is quite difficult to derive the timing from Fig. 4. It will be much easier if the range of timing is provided in a supplementary table.

References:

Alerstam, T., and Å. Lindström. 1990. Optimal Bird Migration: The Relative Importance of Time, Energy, and Safety. In Bird Migration, 331–51. Springer Berlin Heidelberg.

Jacoby, D. M., & Freeman, R. 2016. Emerging network-based tools in movement ecology. Trends in Ecology & Evolution, 31(4), 301-314.

Lank, David B., Robert W. Butler, John Ireland, and Ronald C. Ydenberg. 2003. Effects of Predation Danger on Migration Strategies of Sandpipers. Oikos 103 (2): 303–19.

Verhoeven, M.A., Loonstra, A.H.J., McBride, A.D., Macias, P., Kaspersma, W., Hooijmeijer, J.C.E.W., van der Velde, E., Both, C., Senner, N.R. and Piersma, T. (2020), Geolocators lead to better measures of timing and renesting in black-tailed godwits and reveal the bias of traditional observational methods. J Avian Biol, 51

Reviewer #3: In their study, Lagasse and colleagues provide a flyway network model based on tracking data from different breeding populations and subspecies of Dunlins migration through the East Asian-Australasian Flyway. Using the network approach, the authors characterized migration strategies (routes, phenology) and compared the different breeding populations and subspecies. First, I would like to congratulate all authors on collecting such an impressive collection of tracks from this species and for putting them together allowing to compare the subspecies and reveal some of the larger scale patterns of migration in the EAAF. The paper is very well written and easy to understand. The methods include one of the most elaborate and transparent description of light level geolocation I have seen and the work the authors put into the track estimates allowing robust conclusions. While reading the paper, I of course though that information on population dynamics would make the study so much more valuable. I am aware that that information is not yet available (and that there are plans already installed in will likely be installed in the future to get these urgently needed information). The lack of these data is discussed in the “Limitations and future directions” section, and I am sure that the network developed in this study can be used on the future and may even can inform effective monitoring plans.

In my opinion, the paper is ready to be published. If I can suggest a slight improvement, it would be to also put some of the values presented in the tables into figures (e.g. departure/arrival/wintering duration etc. for the different populations). I think that this would make it easier to see the differences/communalities. I got a (tiny) but frustrated while trying to extract and understand the dates/periods and picture them myself in the head. I ended up visualizing them for myself to get a better impression.

Two additional very minor comments:

L95: Sounds like migration only presents risks. Maybe add that migrations evolved due to the opportunity to exploit optimal conditions across regions.

L124: What about Morrick et al. 2022 Conservation Science and Practice

Again, congratulations on the data collection and the synthesizing analysis across populations. I am sure that this study is of interest to the migration/shorebird community and that the methods described will be appreciated and repeated in future studies.

Simeon Lisovski,

Alfred Wegener Institute, Potsdam

6. PLOS authors have the option to publish the peer review history of their article (what does this mean?). If published, this will include your full peer review and any attached files.

Reviewer #1: **Yes: **Chi-Yeung Choi

Reviewer #2: No

Reviewer #3: **Yes: **Simeon Lisovski

---

## [Author Response · Author response to Decision Letter 0]

17 Jun 2022

See attached cover letter and response to reviewers

---

## [Editor Report · Decision Letter 1]

22 Jun 2022

Migratory network reveals unique spatial-temporal migration dynamics of Dunlin subspecies along the East Asian-Australasian Flyway

PONE-D-22-05779R1

Dear Dr. Lagasse,

We’re pleased to inform you that your manuscript has been judged scientifically suitable for publication and will be formally accepted for publication once it meets all outstanding technical requirements.

Kind regards,

Peng Chen, Ph.D.

Academic Editor

PLOS ONE
---

## [Editor Report · Acceptance letter]

27 Jul 2022

PONE-D-22-05779R1 

Migratory network reveals unique spatial-temporal migration dynamics of Dunlin subspecies along the East Asian-Australasian Flyway 

Dear Dr. Lagassé:

I'm pleased to inform you that your manuscript has been deemed suitable for publication in PLOS ONE. Congratulations! Your manuscript is now with our production department. 

Kind regards, 

on behalf of

Dr. Peng Chen 

Academic Editor

PLOS ONE